# Effect of concussion on salary and employment: a population-based event time study using a quasi-experimental design

Peter Fallesen,[1,2] Benito Campos [iD] [3,4]

[1]Swedish Institute for Social Research, Stockholm University, Stockholm, Sweden
[2]ROCKWOOL Foundation, Copenhagen, Denmark
[3]Department of Neurosurgery, Heidelberg University Hospital, Heidelberg, Germany
[4]Department of Health Tech, Technical University of Denmark, Lyngby, Denmark

**Correspondence to**
Dr Peter Fallesen;
peter.fallesen@sofi.su.se

## ABSTRACT

**Objective** Concussions are the most frequent traumatic brain injuries. Yet, the socioeconomic impact of concussions remains unclear. Socioeconomic effects of concussions on working-age adults were studied on a population scale.

**Design** This population-based, event time study uses administrative data as well as hospital and emergency room records for the population of Denmark.

**Setting** We study all Danish patients, aged 20–59 years, who were treated at a public hospital or at an emergency room between 2003 and 2017 after suffering a concussion without other intracranial or extracranial injuries (n=55 424 unique individuals). None of the patients had a prior diagnosis of intracranial or extracranial injuries within the past 10 years leading up to the incident.

**Primary and secondary outcome measures** As primary endpoint, we investigate the mean effect of concussion on annual salaried income within a 5-year period after trauma. In an exploratory analysis, we study whether the potential impact of concussion on annual salaried income is driven by patient age, education or economic cycle.

**Results** Concussion was associated with an average change in annual salary income of −€1223 (95% CI: −€1540 to −905, p<0.001) corresponding to a salary change of −4.2% (95% CI: −5.2% to −3.1 %). People between 30 and 39 years and those without high school degrees suffered the largest salary decreases. Affected individuals leaving the workforce drove the main part of the decrease. Absolute annual effect sizes were countercyclical to the unemployment rate.

**Conclusions** Concussions have a large and long-lasting impact on salary and employment of working-age adults on a nationwide scale.

## INTRODUCTION

Concussions are by far the most frequently occurring intracranial injuries affecting approximately 450–600 per 100 000 people every year.[1] Immediate symptoms may last for days or weeks. Further, Danish cohort data[2] indicate that 10%–15% of patients diagnosed with concussion suffer from long-term symptoms such as headache, fatigue and intolerance to stress, whereas

### Strengths and limitations of this study

► Natural experiments used to obtain plausible causal effects between concussion and salary/employment.
► Large, population-based sample with multiple data layers.
► Analysis includes how economic cycles affect outcome measures.
► Data only captures concussions registered in emergency rooms and hospitals.
► Because concussions do not occur at random, causal estimate relies on stronger assumptions than for a randomised control trial.

other studies place the upper bound as high as 30%.[3 4] Clinical practice has encouraged patients to restrict social, mental and physical activity in the weeks following a concussion (see Silverberg *et al*[5] for review), although prolonged inactivity may prolong symptoms. Thus, symptoms, comorbidities and suggested treatment are associated with short-term to long-term absence from work and lower productivity.

Yet, the causal effect of concussion on economic burdens for individuals and society through decreased labour market activity has not been identified. First, concussion is a sudden incident and thus not amenable to prospective study nor randomisation. Cohort and case–control studies[6–10] provide some valuable evidence on employment and labour market outcomes among those who suffered concussions but are prone to selection bias. Individuals at high risk of concussions may differ on unobserved characteristics (eg, risk aversion, routine activities) from those at low risk. People who are more likely to suffer concussions may also, on average, have more precarious or unstable employment trajectories prior to the incident, which may further bias prospective studies.

Given the high incidence rate of concussion, even small losses of productivity and discrete drops in employment would have a significant socioeconomic impact and thus, it would require large patient cohorts with suitable controls to grasp the full socioeconomic impact of concussions. Thus, in the absence of the possibility of randomisation, using a natural or quasi-experimental design is the only likely option to parse out the causal effect of concussions on labour market activity.[11]

We examine how concussions affect salary and employment of working-age individuals in Denmark, a representative north-European industrial nation with a strong welfare state and a flexible labour market. We use administrative longitudinal data linked to hospital and emergency room diagnostic data on all Danes, who received a primary diagnosis of concussion between 2003 and 2017. To address the problem of unmeasured bias between those that do and do not experience a concussion, we use a quasi-experimental event-study approach[12 13] where we compare similar individuals, who experienced their concussions at different time points. Under mild assumptions of parallel trends in wage progression prior to concussion and random timing of concussion event within a 5-year time frame, the approach recovers a robust estimation of the effect of concussion on annual salary and employment status.

## MATERIAL AND METHODS
### Data sources and sample construction
Concussion data originate from the Danish National Patient Registry (DNPR) (see Schmidt *et al*[14] for description). DNPR is published annually and holds information on all hospitalisations at public hospitals in Denmark since 1977, on all emergency room visits and outpatient treatments at public hospitals since 1994 and almost all private hospital treatments since 2003. With one single, short-lived exception, private hospitals do not operate emergency rooms in Denmark. Since 2003, the data cover 95% of all treatments at private hospitals,[14] yet only 13 concussions were diagnosed in private hospital settings throughout the period covered by the data.

The combined exposure and control cohort includes all Danes aged 20–59 years, who received a primary diagnosis of concussion (International Classification of Diseases (ICD) 10th revision code S06.0, ICD-8 code N850) between 2003 and 2017 and did not sustain any kind of additional intracranial or extracranial injury. Individuals who regularly engage in activities associated with a high risk of sustaining multiple concussions may differ from the average concussion patient and would likely be over-represented in the exposure sample. To avoid such potential bias, all individuals who were diagnosed with any kind of brain trauma during a 10-year period prior to the concussion event were excluded. Altogether, the study included a cohort of 55 424 individuals. Only attrition is through mortality and out-migration, and out-migrated or deceased patients with missing spells in the follow-up period are excluded in those periods.

As a measure of productivity, a price-index deflated annual salaried income was used. Salary information comes from Statistics Denmark's Income Statistics Database. The database includes all declared annual incomes including income from self-employment. The Danish Tax Authorities supply the data to Statistics Denmark. Overall accuracy is considered very good.[15] Table 1 reports number of observations for the samples and number of observations with missing salary information. As evident, only between 0.01% and 0.02%t of observations across exposure and control groups have missing salary information. These observations were

**Table 1** Number of observations for exposure and control groups across time since exposure and number of observations with missing salary information

| Years until exposure | Exposure group | Control Δ=1 | Control Δ=2 | Control Δ=3 | Control Δ=4 | Control Δ=5 |
|---|---|---|---|---|---|---|
| −4 | 36 804 | 33 681 | 31 112 | 29 190 | 27 859 | 26 794 |
| −3 | 36 978 | 33 834 | 31 245 | 29 366 | 27 973 | 26 907 |
| −2 | 37 195 | 34 003 | 31 407 | 29 501 | 28 146 | 27 031 |
| −1 | 37 449 | 34 224 | 31 582 | 29 687 | 28 288 | 27 220 |
| 0 | 37 848 | 34 551 | 31 851 | 29 922 | 28 530 | 27 421 |
| 1 | 37 467 | | 31 755 | 29 832 | 28 433 | 27 337 |
| 2 | 36 940 | | | 29 807 | 28 421 | 27 295 |
| 3 | 36 484 | | | | 28 421 | 27 304 |
| 4 | 36 084 | | | | | 27 314 |
| Total observations | 333 249 | 170 293 | 188 952 | 207 305 | 226 071 | 244 623 |
| Observations with missing salary | 81 | 32 | 31 | 44 | 35 | 29 |

Control groups have not suffered a concussion in 10+Δ years before incident; exposure group has not suffered concussion the 10 years before exposure incident.

disregarded in the main analysis. Through social security numbers, information on salaried income were linked to records on diagnosed concussions. Further, information on high school or equivalent level degree at time of concussion was obtained using the Danish Education Database. The Danish Population Database provided demographic information on age and gender for all respondents. Since the data used in the study come from deidentified administrative registers that Statistics Denmark makes available for research purposes for approved institutions, no approval from an ethics committee was needed to carry out the study. The research was carried out as part of project no. 706 630 approved by Statistics Denmark. Statistical analysis was carried out using Stata MP V.15.1.

## Quasi-experimental design

The study used a quasi-experimental, difference-in-differences event time approach previously described in a health setting by Dobkin *et al.*[12] The approach compares two groups of individuals from the same cohort, where both groups experience concussions, but at two different time points ($t_c$, $t_c+\Delta$). Specifically, the sample of 55 496 individuals was divided into six different subgroups: (1) the exposure group, which includes all patients who suffered their concussion during the period 2003–2012 (n=37 848) and (2) five control groups, which comprise patients who experienced their concussions $\Delta$={1 (n=34 551), 2 (n=31 851), 3 (n=29 922), 4 (n=28 530) and 5 (n=27 421)} years later than the exposure group and did not experience any kind of brain injury in the 10+$\Delta$ years before the concussion event (note that the design allows individuals to both be part of the exposure and control groups). The model is built on the assumption that the exact timing of a concussion is random for small enough values of $\Delta$, and on the additional assumption that the exposure and the control groups would have displayed parallel trends in salary if the control group had not suffered a concussion at $t_c$ (ie, assuming that control and exposure group(s) would have continued to show similar trends in salaried earnings had the exposure group not experienced concussions). Table 1 shows the number of patients in the exposure group and the five control groups for each year relative to exposure group's concussion incident. Using multiple comparison groups makes it possible to gage the validity of the assumption that the exact timing of a concussion is random for small enough sizes of $\Delta$.

To estimate the impact of concussion on labour market outcomes, the analysis focuses on the change in annual salary as the primary outcome, and, in further exploratory analyses, studies additional outcomes such as income from health-related benefits, income from welfare benefits and employment rates. The data are nested within a three-level structure: exposure or control group $g$, which includes individuals $i$, at times to exposure groups' concussion incident $t$. First, a

standard difference in differences model for each separate control group $\Delta$={1, 2, 3, 4 and 5} is estimated using ordinary least squares:

$$Salary_{git} = \beta_0 + \gamma Exposure_g + \theta Post_t + \delta Post \times$$
$$Exposure_{git} + X_i\beta + \sum_{Age=20}^{59+\Delta} I(Age)\,\eta_{Age} + \quad (1)$$
$$\sum_{Year=1999}^{2012} I(Year)\,\eta_{Year} + \epsilon_{git}$$

where $Salary_{git}$ measures annual salaried income adjusted to 2015-prices using the consumer price index; $Exposure_g$ indicates whether the observation belongs to the exposure or control group; $Post_t$ captures the period after the exposure group's concussion occurred; $Post_t \times$-$Exposure_{git}$ captures the effect concussion, measured as share of year t≥0 affected by concussion (ie, for year of incident exposure is expressed as share of year spent with post–exposure, for following years it is equal to 1); $X_i$ is a set of covariates that includes a high school indicator and a gender dummy; $\epsilon_{git}$ is the error term and the two last sets of indicator variables $I(Age)$ and $I(Year)$ capture age and incident year (for control group, the year indexed against). Under an assumption of parallel trends in salaried earnings (ie, assuming that control and exposure group(s) would have continued to show similar trends in salaried earnings had the exposure group not experienced concussions), $\delta$ then captures the annual causal effect of concussion on salary for people exposed to concussions (see online supplemental methods for further details). For additional exploratory analyses, separate models across gender, educational level and age, as well as across the salary distribution are also estimated (see online supplemental methods, online supplemental digital content 1, for further details). The authors document and make available all code needed to reproduce the findings in the study (online supplemental digital content 2).

## Standard protocol approvals, registrations and patient consents

Since the data used in the study come from deidentified administrative registers that Statistics Denmark makes available for research purposes for approved institutions, no approval from an ethics committee was needed to carry out the study. The research was carried out as part of project no. 706 630 approved by Statistics Denmark.

## Patient and public involvement

There was no involvement from patients or members of the public in the design, or conduct, or reporting, or dissemination plans of the research.

## RESULTS
## Concussion leads to long-term loss in salaried income

Individuals who suffered a concussion (exposure group) had a lower salaried income compared with individuals who experienced their concussion 1–5 years after

**Table 2** Estimated effect of concussion on salary of exposure group compared with control groups that suffered their concussion Δ=1, 2, 3, 4 and 5 years after the exposure group's concussion event, measured at 2015-level. $N_{Exposure}$: 37 848

|  | Estimated salary effect (δ) | 95% CI | P value | $N_{Control}$ |
|---|---|---|---|---|
| Δ=1 year | −€423 | (−€919 to 73) | 0.095 | 34 551 |
| Δ=2 years | −€825 | (−€1108 to −543) | <0.001 | 31 851 |
| Δ=3 years | −€1019 | (−€1331 to −707) | <0.001 | 29 922 |
| Δ=4 years | −€1126 | (−€1446 to −805) | <0.001 | 28 530 |
| Δ=5 years | −€1243 | (−€1564 to −922) | <0.001 | 27 421 |

Results obtained from estimations following Eq. (1). Models include controls for high school diploma, gender, age and observation year. Results obtained using reghdfe in Stata.

the exposure group (control groups). Compared with patients who experienced a concussion 1 year after the exposure group, salaried income was €423/£380 (95% CI: −€9129/−£8208 to 73/66, p=0.095; table 2) lower, corresponding to a salary decrease of 1.5% (95% CI: −0.3% to 3.2%; figure 1). Compared with patients who experienced a concussion 5 years after the exposure group, however, salaried income in the exposure group was €1243 (95% CI: −€1564/−£1406 to −922/−829,

p<0.001) lower, corresponding to a salary decrease of 4.2% (95% CI: 3.1% to 5.3%; figure 1). Normalised wage progression for the control groups, who suffered a concussion 1–5 years after the exposure group, showed similar trends and similar levels pre exposure, indicating that the parallel wage trends assumption was met (figure 2 and online supplemental table S1 and online supplemental figure S1 in online supplemental digital content 3).

We hypothesised that the salary decreases resulted from a combination of lower salary and exit from the labour market, either through short-term or long-term absence/unemployment. In an exploratory analysis, we tested whether labour force exit drove the full effect of concussion on salary (figure 3). By comparing the cumulative distribution of salary density for the exposure group with the cumulative distribution of salary density for the Δ=5 control group (figure 3, left panel), we found that the impact of concussion on salary was significant for individuals in the lower quartile of the salary distribution (at a 95% significance level). Specifically, below a threshold salaried income of €40 000 (£36 000), the presumed impact of concussion on salary increased towards the bottom of the earnings pyramid; figure 3, right panel .

Comparing the exposure group to the control group Δ=5, which suffers a concussion 5 years after the exposure group, concussion was associated with a 2.6% (95% CI:

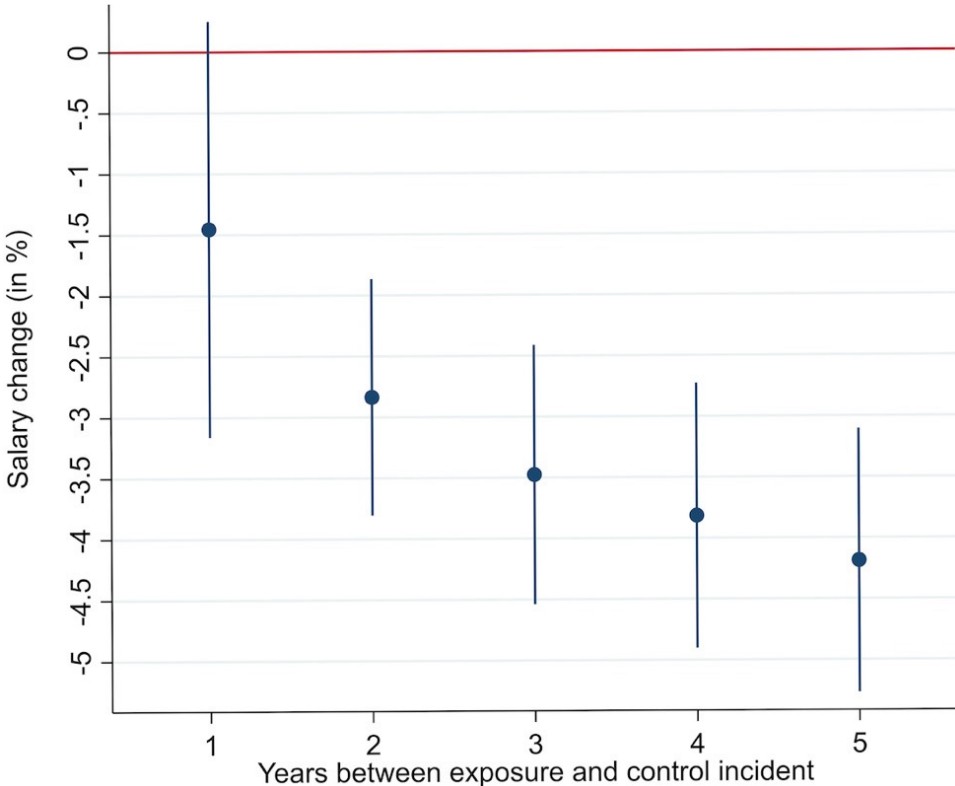

**Figure 1** Estimated effect of concussions in percentage on salary for the exposure group measured against each control group. Note: figure shows the percentage change in salary experienced by the exposure group following their concussions compared with the expected trajectory absent the concussion (calculated from the control groups) with 95% CIs. See table 1 for separate p values for each estimate.

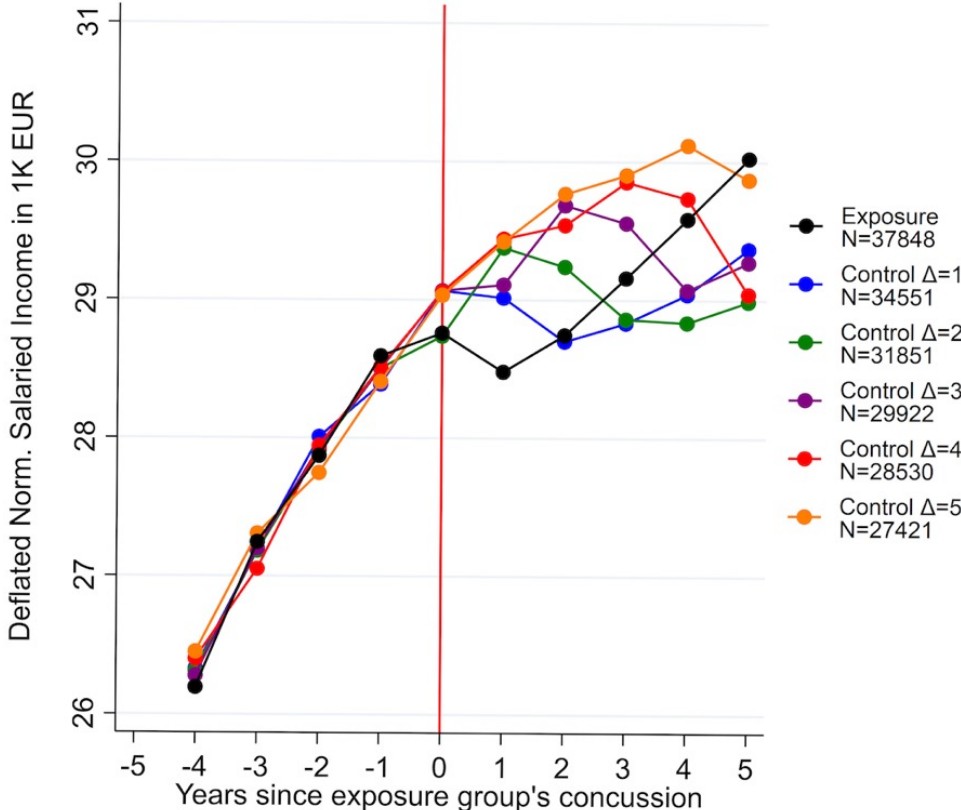

**Figure 2** Salary development for exposure and control groups across time of exposure. Note: figure shows the salary trajectories for the exposure group (black) who suffers concussion at year 0 against normalised wage trajectories for the control groups who suffer their concussions 1–5 years later. Δ indicates the number of years between exposure and control incident. Table shows that there are no significant differences in the normalised salary levels for exposure and control group prior to exposure incident (see online supplemental figure S1, online supplemental digital content 3 for unnormalised salary trajectories).

3.0% to 2.2 %, p<0.001) increase in the risk of receiving €0 in annual salary.

### Long-term loss in salaried income stems from exit from the labour market

To further examine whether exit from the labour market was caused either through short-term or long-term absence/unemployment, we estimated a dynamic model using the control group Δ=5, which suffers a concussion 5 years after the exposure group. Sick leave benefits payments were higher in the exposure group compared with the control groups for the first 2 years following concussion. Sick leave benefits were no longer different from year 3, while the difference in annual salary between exposure and control groups persisted. Further, employment in the exposure group remained lower than in the control group Δ=5 and remained so for the entire postexposure period (see online supplemental table S2, online supplemental digital content 3 for further details). To assess whether some form of public benefits covered part of the salary loss, total income decline was compared with salary decline following concussion. Indeed, total income decline was lower than the salary decline through a 5-year period (see online supplemental figure S2, online supplemental figure 3 for further details).

### Younger patients without high school degree drove the effect of concussion on income loss

The exposure group and all control groups differed slightly in terms of average patient age, male-to-female ratio and for control group Δ=5, in the frequency of individuals with at least a high school degree (see online supplemental table S3, online supplemental figure 3 for further details). To ensure that differences in gender, education or age did not influence our results, we subdivided our exposure group into subgroups based on gender, education status and age at time of concussion. We then estimated the impact of concussion on salary and employment across all values of Δ and for all subgroups (see, online supplemental figures S3–S8, online supplemental figure 3 for further details). Patients between age 30 and 39 years and those without a high school degree experienced the largest absolute and relative declines in salary.

Finally, we addressed the role of timing of concussion across different years. Given that per design our exposure group always suffered their concussion earlier than the control groups do, changing labour market conditions could moderate effects. Part of our sample suffered its concussion during or just prior to the Great Recession in 2009 and 2010, which arguably presented the largest

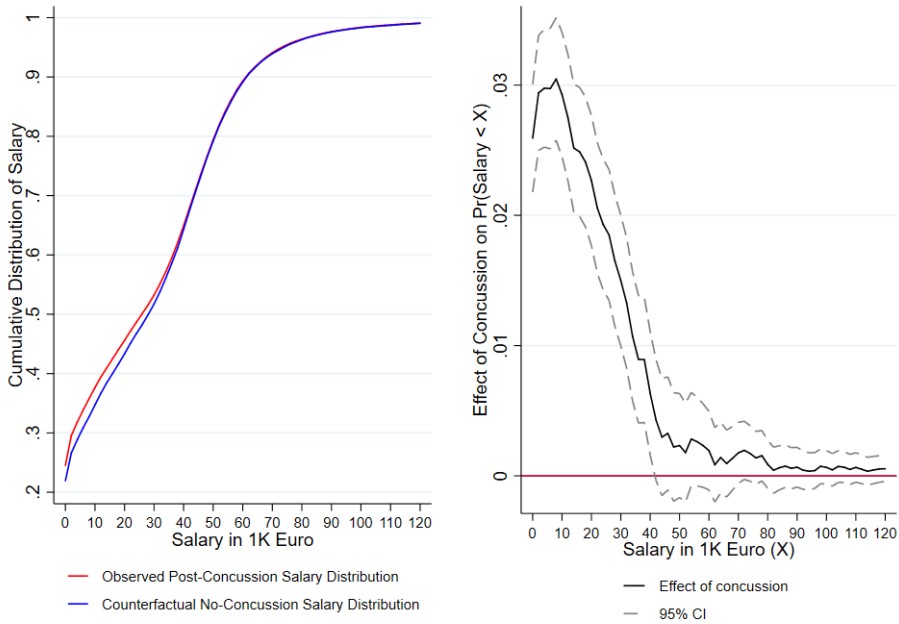

**Figure 3** (Left panel) The cumulative density function for salary post treatment among the treatment groups and their counterfactual outcome had they not experienced their concussions, and (right panel) the change in salary density for the exposure group compared with their counterfactual baseline expressed as the effect of concussion on the probability of earning below the salary level expressed on the x-axis following exposure event. Note: the figure shows the observed cumulative salary distribution following concussion for the exposure group (red) and the expected counterfactual salary distribution absent suffering concussion in the exposure group (blue), when using Δ=5 control group. The black line shows the difference between the observed and the counterfactual distribution, and the grey dash lines show the 95% CI. The close to constant decline of the difference between the two distributions as the salary increase indicates that the main part of the effect of concussions on salary is driven by people having a salary equal to 0.

shock to both the global and local economy since the Great Depression in the 1930s. In Denmark, the great recession was preceded by a series of years of economic growth, low unemployment and increasing salaries (see online supplemental figure S8, online supplemental figure 3 for salary development from 1994 to 2017). We estimated the impact of concussion on salary separately for each year from 2003 to 2012 and plotted the estimate against the percent of full-time unemployment in the Danish labour force (LF) (figure 4). Suffering a concussion during an economic boom had a substantially higher impact on salary than doing so during a recession when comparing to control groups who suffered concussions 2–5 years later than exposure group.

## DISCUSSION

The impact of concussion on employment and salary remains understudied. In a systematic review of four studies on the association between mild traumatic brain injury (TBI) and return to work, Cancelliere et al[9] found that most workers return to work within 3–6 months of suffering a mild TBI, but that the long-term impact (more than 1 year after concussion) was not studied. In addition, studies included small-to-medium sample sizes, varied measures of return to work and employed

both case–control and cohort designs. Using an inception cohort study design, Theadom et al[7] collected follow-up data 4 years after mild TBI incidents on 245 New Zealanders who were employed prior to incident. They found a 3.6% productivity decline among those who suffered a concussion, compared with a 2.3 population average decline. The group suffering mild TBI also reported more difficulties carrying out work-related tasks. In a related study,[7] Theadom et al further found that the mild TBI group had persisting cognitive symptoms 4 years after suffering their concussion compared with an age-sex matched control group. Also, using a case–control design and data from Taiwan, Chu et al[6] found that 1 month after incident, 26% of patients had still not managed to return to work, and a large share of those who did return scored below full-time employment on a work quality index. Only one other study by Graff et al[10] included a large patient cohort (n=19 732). Using case–control, they not only found an association between concussion and failing to return to work of 1.54 OR but also found that exposed individuals had lower labour market attachment and were more likely to receive health related benefits pre incident compared with the control group.

In the present study, we overcame some of the obstacles faced by previous work on the impact of concussions

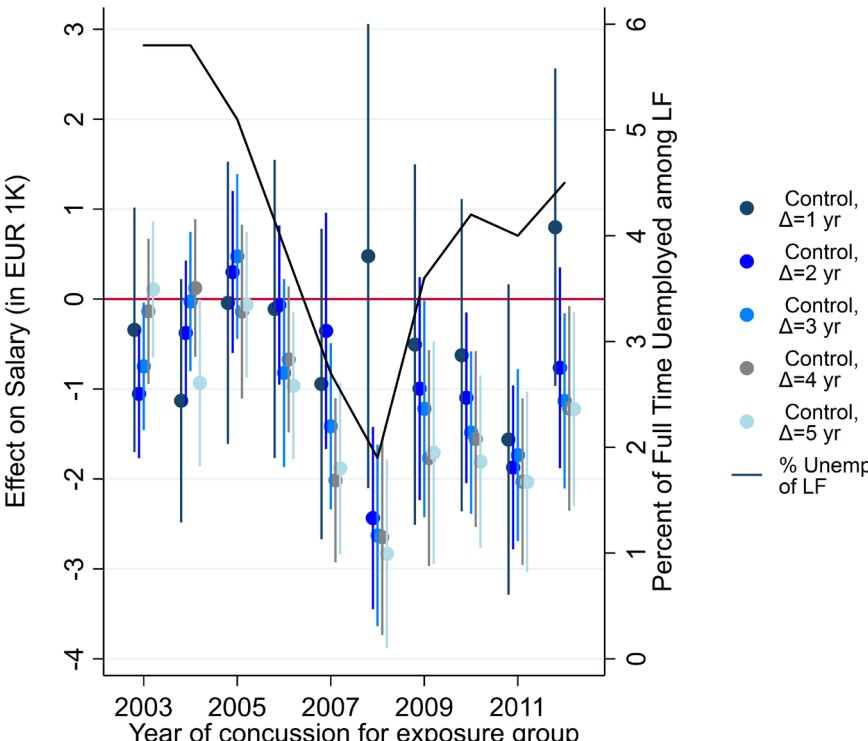

**Figure 4** Effect of concussion on salary across incident years and control groups together with the percentage fulltime unemployed of the labour force (LF). Note: figure shows annual estimates of concussion against each control group separately mapped against the share of the labour force that is full-time unemployed. 95% CIs. The estimates for the effect of concussion on salary almost uniformly increase in absolute magnitude when unemployment decreases, and decrease when unemployment increases, indicating that the effect of concussion on salary is countercyclical to the economic cycle.

on labour market outcomes by including a vast cohort of patients and exploiting a quasi-experimental design that allows us to plausibly account for unobserved difference between exposure and control groups. In such a quasi-experimental setup, exposure and control groups only differ in the timing of concussion. Since everyone in the control group experiences a concussion within 5 years after individuals in the exposure group, the groups are likely to be balanced on unobservable characteristics. This is particularly important given the number of potential factors that can influence employment after concussion.[16 17] Data from Donker-Cools et al,[17] for instance, suggest larger employers are more able to keep those who have sustained brain injuries in work compared with smaller employers. Furthermore, since our data did not include health-related data such as past psychiatric history, we cannot exclude that exposure and control groups differed in health-related aspects and that these differences biased our results, that is, that an individual left the workforce for a concussion-unrelated cause like a psychiatric disease triggered by the stress of a concussion event. Thus, even if we believe that our quasi-experimental setup leaves us with exposure and control groups that only differ in the timing of concussion, especially given the reported sample sizes and the finding that exposure and control groups show similar pre-exposure trends

on both primary and secondary outcomes (figure 1 and online supplemental tables S1 and S2), this aspect needs to be discussed as a potential limitation of our study.

In addition, salary and employment data reported here were compiled routinely through third-party reporting and were mandatory for all subjects, thus giving a complete and comprehensive picture of the economic impact of concussion on a nationwide scale. It should be mentioned that our study also included data from individuals diagnosed in private hospitals. However, given the setup of the Danish healthcare system, that is, private hospitals predominantly do selective and overflow surgery and have no ERs,[18] only 13 patients were diagnosed at a private hospital from 2003 onwards.

Altogether, we showed that Danes between 20 and 59 years of age who suffered a concussion during the period 2003–2012 experienced average salary losses of 4.2%. The impact of concussions on salary already materialised 1 year after the incident and remained sizeable for at least 5 years. This result is in line with a 'burn-in' period in which the impact of concussion on wages fully develops. First, concussions occur at some point during the year, thereby not affecting already earned salary that year. Second, in Denmark, most employees are entitled to receive their salary during sickness absence for an open ended, but not indefinite period. The results further showed that both in absolute and relative terms,

people with an educational level at less than a high school degree saw substantially larger negative impact to salaried earnings than did those with at least a high school degree. Also, the group with less than a high school degree also saw an immediate impact on salary from their concussion (cf. online supplemental figure S4), indicating that the burn-in period present for workers with at least high school education likely expressed differences in types of employment and job protection.

In addition, total income decline was lower than the salary decline through a 5-year period (see online supplemental figure S2, online supplemental figure 3 for further details), suggesting that the impact of concussions on salary largely stems from affected individuals leaving the labour force completely, likely sustaining themselves through early retirement, disability pensions, self-sufficiency or other income sources instead.

It is important to mention that our study was restricted to individuals diagnosed in ER and hospital settings and individuals diagnosed by a general practitioner (GP) might differ from the population studied here. Rowson *et al*, however, show that in concussed individuals, severity of the cranial injury is not strongly correlated with strength or length of subsequent symptoms.[19] Thus, individuals diagnosed by a GP might suffer concussion effects as much as individuals who initially sustained a more severe cranial injury and sought medical attention in an ER or hospital setting. If this holds true, our results may have validity beyond individuals diagnosed in an ER or hospital setting.

If we assume that people return to their expected salary levels after a 5-year recovery period (a very conservative assumption that is not supported by our data), the mere net annual salary loss in our sample would be approximately €23 000 000 (£21 000 000) measured in 2015-value. That would neither include hospital charges, medical costs for the treatment of concussion, the foregone tax from income and the increased need for welfare spending, nor would it account for the large group of individuals who never seeks treatment[20] or receive their diagnosis from their general practitioner rather than in a hospital or emergency room, and thus escape our study. Thus, total public costs are likely substantially higher.

In addition, both in absolute and relative terms, the early peak-working-aged individuals (30–39 years) and the less-educated individuals in our cohort seemed to be most affected after suffering a concussion. These findings might have an additional and yet unmeasured social impact, especially if our results are transferrable to other nations with a less established welfare state and a less flexible labour market. In such countries, the impact on the young and less-educated individuals suffering a concussion and thus on society might be accentuated.

Comparing our hospital incidence rates to more complete canvases of incidences carried out by Cassidy *et al*,[21] it seems likely that the actual cost in the population is more than twice as large as what we estimate, assuming that individuals not diagnosed in a hospital setting on average suffer the same extent of concussion symptoms. If we were to consider the average concussion incidence rates for six other advanced European countries that are somewhat comparable to Denmark (Norway, Finland, Germany, Netherlands, England and France) and under the assumption that concussion has a similar impact on earnings in these countries, the net annual salary loss would be approximately €1 099 400 000 (£988 4780 000) measured in 2015-value. While our study likely underestimates the total socioeconomic impact of concussion, it suggests that concussions have a large economic impact on a nationwide scale and on productivity and income at the patient level.

## CONCLUSION

Using timing of concussion as a natural experiment, we provide first plausible causal estimates of the effect of concussion on salary and employment among patients treated for concussion in an emergency room or hospital setting in Denmark, 2003–2017. Our results show that among this patient group, concussion has a large and long-term negative causal impact on salary and employment. People between 30 and 39 years and those without high school degrees suffered the largest salary decreases.

**Acknowledgements** The authors thank Torben Heien Nielsen for helpful comments on an earlier draft of the manuscript.

**Contributors** PF and BC conceived of the presented idea. PF performed the computations. PF and BC verified the statistical methods. PF and BC discussed the results and wrote the manuscript. The corresponding author confirms that he had full access to all the data in the study and had final responsibility for the decision to submit for publication.

**Funding** This work was supported by the Swedish Research Council for Health (PF), Working Life and Welfare (Grant no. 2016-07099) with additional funding from the ROCKWOOL Foundation (PF). The research was carried out independently of the funders.

**Competing interests** None declared.

**Patient consent for publication** Not required.

**Provenance and peer review** Not commissioned; externally peer reviewed.

**Data availability statement** Data may be obtained from a third party and are not publicly available. The data used in this study have been made available through a trusted third party, Statistics Denmark. Due to privacy concerns, the data cannot be made available outside the hosted research servers at Statistics Denmark. University-based and private Danish scientific organisations can be authorised to work with data within Statistics Denmark. Such organisation can provide access to individual scientists inside and outside of Denmark. Requests for data may be sent to Statistics Denmark: http://www.dst.dk/en/OmDS/organisation/TelefonbogOrg.aspx?kontor=13&tlfbogsort=sektion or the Danish Data Protection Agency: https://www.datatilsynet.dk/english/the-danish-data-protection-agency/contact/. The authors document and make available all code needed to reproduce the findings in the study.

**ORCID iD**
Benito Campos http://orcid.org/0000-0001-7243-8333

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
