## [Reviewer comments · BMJ Open]

ARTICLE DETAILS

TITLE (PROVISIONAL)	The Effect of Concussion on Salary and Employment-A Population-Based Event Time Study using a Quasi-Experimental Design
AUTHORS	Fallesen, Peter; Campos, Benito

VERSION 1 – REVIEW

REVIEWER	Connor Watkin University of Nottingham, United Kingdom.
REVIEW RETURNED	19-Mar-2020

GENERAL COMMENTS	Overall, this is a really interesting paper. I agree that it is well-needed and there is not a huge amount of evidence regarding the economic impact of mild brain injury. The sample size is very impressive and this is rare in brain injury research due to drop out rates etc. There are just a few things that I would like to query: Introduction: 56: "10-15% of patients diagnosed with concussion suffer from long-term symptoms such as headache, fatigue, and intolerance to stress". Do you have evidence to support this statement? I think it would be useful to have the conversion rate into British pounds as well as Euro's just so that people are able to conceptualise how much of a loss in annual salary this is. Method: 89: You have used first person throughout a lot of the method section. This should be in third person. So for this particular line, it would be: "The concussion data originates from the Danish National Patient registry" as opposed to "our concussion data". This needs amending throughout the methods section. Results: 181, 204, 218: There are quite a few areas where you have discussed the results in the results section. It is generally good practice to only present results in the results section and then analyse these in the discussion section. Discussion/limitations: I think it is important to consider a few more limitations of this research that are not currently mentioned in the discussion. For example, the research can only be generalised to
---

	those who are hospitalised following concussion. Many people who receive mild brain injury's, for example in sport, are not hospitalized. Consequently, we would need further research to investigate if this salary reduction also applies to that group of people. Furthermore, there is evidence available that suggests larger employers are more able to provide accommodations and keep those who have suffered a brain injury in work compared with smaller employers (Donker - Cools, 2016). This may be an additional factor that was not considered in this research that could have influenced why some lost more money than others. Buck (2012) also found that the definition of mTBI is ambiguous and commissioners under-estimate the level of cognitive impairment. This may lead to them over-estimating vocational abilities.
--	--

REVIEWER	Deborah Snell University of Otago New Zealand
REVIEW RETURNED	02-Apr-2020

GENERAL COMMENTS	This is an important and understudied area of the research considering longer term impacts of concussion. The authors have used a large de-identified dataset to examine the socio-economic impacts of concussion over time. The sample includes people of employment age who represented to a hospital or ED after a concussion event. We know a large proportion of people do not present to such services and so are not captured by the study. I think the conclusions need to be revised slightly to be clear that findings may not generalise to all persons experiencing a concussion event. There is an assumption that people leaving the workforce did so because of the concussion event and not some other unrelated reason. For example, past psychiatric history information was not captured so it is not known if people leaving the workforce at some point following a concussion did so because they experienced resurfacing of a pre-existing psychiatric condition following a stressful life event (the concussion) or some other unrelated health issue. I note those with less education and ages 30-39. These groups may be more mobile employment-wise irrespective of concussion. There are some minor areas of the manuscript where the meaning is vague, especially in the results section. In places it is unclear if rates discussed are annual rates and which groups and outcomes are being compared is unclear at times. For example the statement on page 12 (lines 194-196) suggests sick leave benefits payments were higher in the exposure group for the first two years following the concussion event. Higher than what? the remaining years? The other groups? In one or two places the authors refer to the concussion incidence - I think they mean incident (e.g. line 63, line195). There is an error in reference 14. The conclusions need to be revised to avoid overstating the findings.
--

REVIEWER	Wenhui Mao The Center for Policy Impact in Global Health, Duke Global Health Institute, Duke University, USA
REVIEW RETURNED	15-Jun-2020

GENERAL COMMENTS	Since the analysis purely relied on the data from Public hospitals, it would be great to point this out throughout the paper. My assumption is concussion patients using private hospitals may suffer from different income loss. This study extracted multiple information from existing administration databases with great clarity in describing the data extraction and matching process. The model was diligently developed, using different period of patient records as case and control groups. I don't have additional comments regarding methods.
---

REVIEWER	Tyler Lane Monash University Australia
REVIEW RETURNED	22-Jun-2020

GENERAL COMMENTS	This study investigates the effect of concussion on earnings using an interesting quasi-experimental approach. My review focuses on the statistics. Overall the approach appears sound and the study design clever. I have some minor comments.  1. I'm not quite certain the including the year the concussion occurred is the best approach. As the authors acknowledge in the discussion ("...the effect only fully reveal[sp] itself after the first year since exposure incident." "...concussions occur at some point during the year, thereby not affecting already earned salary that year.") From Figure 2 it seems quite obvious that the first year post-concussion has a "halfway" impact on income. I'm not sure if it's possible with this analytical approach, but you may want to consider a phase-in period. Other quasi-experimental study designs like interrupted time series allow this, whereby the time period that includes the event is excluded from analysis. This avoids a mixture of pre- and post-data in a period that is treated as post only that could lead to less precise or underestimated effects. This may also explain why the first-year difference is the only non-significant association, while all others were highly significant at $p < .001$. 2. The plots are often hard to read, and I'm not quite certain what I'm seeing in Figure 3. 3. The authors write "The authors document and make available all code needed to reproduce the findings in the study." It's not quite clear whether this means the code is included (which I can't find), or interested parties must contact the authors. My preference would be to include it with the manuscript, and to also name the statistical software they used.
--

REVIEWER	Yana Ma Soochow University, China.
REVIEW RETURNED	28-Jun-2020

GENERAL COMMENTS	The reviewer provided a marked copy with additional comments. Please contact the publisher for full details.
--

VERSION 1 – AUTHOR RESPONSE

We thank all reviewers for the positive evaluation of our study.

Answers to reviewer 1:

1) “56: “10-15% of patients diagnosed with concussion suffer from long-term symptoms such as headache, fatigue, and intolerance to stress”. Do you have evidence to support this statement?”

Data was taken from: Pinner M, Jensen R, Birket-Smith M, Gade A, Riis JO, others:

Konsensusrapport om Commotio Cerebri (Hjernerystelse) [Consensus Report on Commotio Cerebri (Concussion)]. Videnscenter for Hjernesgade, 2003. We have added the reference and cite additional, pertinent international studies to back up our claim.

2) “I think it would be useful to have the conversion rate into British pounds as well as Euro's just so that people are able to conceptualise how much of a loss in annual salary this is.”

We have converted figures to British pounds, following the conversion rate €1 = £0.9.

3) “89: You have used first person throughout a lot of the method section. This should be in third person. So for this particular line, it would be: “The concussion data originates from the Danish National Patient registry” as opposed to “our concussion data”. This needs amending throughout the methods section.”

We have amended the text accordingly.

4) “181, 204, 218: There are quite a few areas where you have discussed the results in the results section. It is generally good practice to only present results in the results section and then analyse these in the discussion section.”

We have edited the text accordingly and confined discussion of results to the discussion section.

5) “Discussion/limitations: I think it is important to consider a few more limitations of this research that are not currently mentioned in the discussion. For example, the research can only be generalised to those who are hospitalised following concussion. Many people who receive mild brain injury's, for example in sport, are not hospitalized. Consequently, we would need further research to investigate if this salary reduction also applies to that group of people. ”

This point is well taken. We now highlight, both in the first paragraph of the materials and methods section as well as in the discussion, that our result's external validity is restricted to individuals diagnosed in an ER or a hospital setting as well as to non-hospitalized individuals, who were diagnosed in the ER but were not admitted. We due, however, also link our findings to Rowson et al, 2018 J Neurotrauma, who show that in concussed individuals, severity of the cranial injury is not strongly correlated with strength or length of subsequent symptoms. Thus, individuals diagnosed by a GP might suffer concussion effects as much as individuals who initially sustained a more severe cranial injury and sought medical attention in an ER or hospital setting. If this holds true, our results may have validity beyond individuals diagnosed in an ER or hospital setting.

6) “Furthermore, there is evidence available that suggests larger employers are more able to provide accommodations and keep those who have suffered a brain injury in work compared with smaller employers (Donker - Cools, 2016). This may be an additional factor that was not considered in this research that could have influenced why some lost more money than others. Buck (2012) also found that the definition of mTBI is ambiguous and commissioners under-estimate the level of cognitive impairment. This may lead to them over-estimating vocational abilities.”

We thank the reviewer for bringing to our attention the studies by Donker-Cools et al. and by Buck et al.. We have incorporated them as part of our discussion.

Answers to reviewer 2:

1) "This is an important and understudied area of the research considering longer term impacts of concussion. The authors have used a large de-identified dataset to examine the socio-economic impacts of concussion over time. The sample includes people of employment age who represented to a hospital or ED after a concussion event. We know a large proportion of people do not present to such services and so are not captured by the study. I think the conclusions need to be revised slightly to be clear that findings may not generalise to all persons experiencing a concussion event. There is an assumption that people leaving the workforce did so because of the concussion event and not some other unrelated reason. For example, past psychiatric history information was not captured so it is not known if people leaving the workforce at some point following a concussion did so because they experienced resurfacing of a pre-existing psychiatric condition following a stressful life event (the concussion) or some other unrelated health issue."

We thank the reviewer for the positive evaluation of our study and for the valuable comment. We have discussed the limitations of our results accordingly. Specifically, we stress that our results only cover individuals diagnosed in ER and hospital settings. In addition, we discuss in more detail why we believe that our quasi-experimental setup leaves us with exposure and control groups that only differ in the timing of concussion. Arguably, because everyone in the control group experiences a concussion within five years after individuals in the exposure group experienced their concussions, the groups are likely to be balanced on unobservable characteristics, especially given the reported sample sizes.

2) "I note those with less education and ages 30-39. These groups may be more mobile employment-wise irrespective of concussion"

We agree with the reviewer that these groups may be more mobile employment-wise irrespective of concussion. To ensure that differences in education or age did not influence our results, we subdivided our exposure group into subgroups based on education status and age at time of concussion. We then estimated the impact of concussion on salary and employment again and across all values of Δ and for all subgroups (see, Figures S3-S8, Supplemental Digital Content 2 for further details). For example, we studied the impact of concussion on salaried income for the subgroup of individuals with less education. In this subgroup all individuals had less than high school degree, differing only in the timing of concussion. Altogether, we show that patients between age 30-39 and those without a high school degree experienced the largest absolute and relative declines in salary.

3) "There are some minor areas of the manuscript where the meaning is vague, especially in the results section. In places it is unclear if rates discussed are annual rates and which groups and outcomes are being compared is unclear at times. For example the statement on page 12 (lines 194-196) suggests sick leave benefits payments were higher in the exposure group for the first two years following the concussion event. Higher than what? the remaining years? The other groups?"

We have re-written the text accordingly to enhance clarity.

Answers to reviewer 3:

1) "Since the analysis purely relied on the data from Public hospitals, it would be great to point this out throughout the paper. My assumption is concussion patients using private hospitals may suffer from different income loss."

We thank the reviewer for his important observation. We apologize for omitting that our data do in fact include concussions diagnosed in private hospitals from 2003 onwards. However, given the setup of the Danish health care system, i.e. private hospitals predominantly do selective and overflow surgery and have no ERs (Denmark has had one private ER, which was in operation for less than a year), we only count 13 patients diagnosed at private hospitals from 2003-2017. We present this data in more detail in the results and in the discussion sections. We also discuss that our data does not include

diagnoses obtained from GPs which in turn may limit the validity of our findings for this patient population.

Answers to reviewer 4:

1) "I'm not quite certain the including the year the concussion occurred is the best approach. As the authors acknowledge in the discussion ("...the effect only fully reveal[sp] itself after the first year since exposure incident." "...concussions occur at some point during the year, thereby not affecting already earned salary that year.") From Figure 2 it seems quite obvious that the first year post-concussion has a "halfway" impact on income. I'm not sure if it's possible with this analytical approach, but you may want to consider a phase-in period. Other quasi-experimental study designs like interrupted time series allow this, whereby the time period that includes the event is excluded from analysis. This avoids a mixture of pre- and post-data in a period that is treated as post only that could lead to less precise or underestimated effects. This may also explain why the first-year difference is the only non-significant association, while all others were highly significant at $p < .001$."

We thank the reviewer for the insightful comment. We do need to point out, however, that our model already incorporates a phase-in period, since the data includes the exact date of concussion diagnosis at the ER/hospital. We now highlight this aspect of our method in the manuscript which is also discussed in more detail in the supplementary materials and methods. Our data further suggest, that the effect of the "burn-in" or "phase-in" is related to differences in employment. As can be seen from Figures S4 and S7, people with less than a high school degree experience a larger salary drop than individuals with at least high school degree. At the same time, however, this effect manifests within the first year following concussion, i.e. without transition period, and remains low during the observation period. We discuss these findings in more detail in results and discussion sections.

2) "The plots are often hard to read, and I'm not quite certain what I'm seeing in Figure 3."

We have divided Figure 3 and its companion Figure S2 into two panels to ease reading. We have also edited the wording of selected figures titles/legends to enhance clarity.

3) "The authors write "The authors document and make available all code needed to reproduce the findings in the study." It's not quite clear whether this means the code is included (which I can't find), or interested parties must contact the authors. My preference would be to include it with the manuscript, and to also name the statistical software they used."

We have added the full code as "Supplementary digital content 3".

Answers to reviewer 5:

1) "Line 47: Please specify the limitations."

We have specified the limitations of our study, both in the strengths and limitations section, as well as in the discussion section.

2) "Line 53: There is no other research in this part. Please add some more."

We have now added additional references and extended the first paragraph of the introduction.

3) "Line 122: Please explain the reasonableness of this grouping."

We have explained the rationale of our approach in more detail in the material and method section.

4) "Line 130: Please explain the rationality of this method."

We have explained the rationality of our method in more detail in the material and method section.

5) "Line 135: Please check the grammar."

We have edited our grammar accordingly.

6) "Line 175: Inconsistent with the data discussed later. Why?"

Thank you for the attention to detail. Due to a rounding error, we have written 4.1 % some places, and 4.2 % other. The correct value is 4.2 % and we have corrected our reported results accordingly.

VERSION 2 – REVIEW

REVIEWER	Connor Watkin University of Nottingham, England
REVIEW RETURNED	24-Aug-2020

GENERAL COMMENTS	I have re-read this paper and compared it with the initial comments that I made back in March. As I initially highlighted, this is a really interesting paper and is well needed for us to fully understand the economic impact of mild brain injury. I am happy that the changes the author has made to this paper make it of an acceptable standard to be published in the BMJ.
--

REVIEWER	Tyler Lane Monash University Australia
REVIEW RETURNED	23-Jul-2020

GENERAL COMMENTS	I thank the authors for their considered response to my review. I have no further comments.
---

REVIEWER	Yana Ma Department of Child and Adolescent Health and Social Medicine School of Public Health, Medical College of Soochow University, China 199 Ren-Ai Road, Suzhou Industrial Park, Suzhou 215123, Jiangsu Province, China
REVIEW RETURNED	03-Aug-2020

GENERAL COMMENTS	The author has made modifications according to the modification Suggestions, and I personally think it is ready for publication. Congratulations to their team.
---

VERSION 2 – AUTHOR RESPONSE

We thank all reviewers once more for the positive evaluation of our study.

Editorials Comments:

“We felt that the following point from reviewer 2 should be discussed as a limitation in your Discussion section:

1) “This is an important and understudied area of the research considering longer term impacts of concussion. The authors have used a large de-identified dataset to examine the socio-economic impacts of concussion over time. The sample includes people of employment age who represented to a hospital or ED after a concussion event. We know a large proportion of people do not present to such services and so are not captured by the study. I think the conclusions need to be revised slightly to be clear that findings may not generalise to all persons experiencing a concussion event.

There is an assumption that people leaving the workforce did so because of the concussion event and

not some other unrelated reason. For example, past psychiatric history information was not captured so it is not known if people leaving the workforce at some point following a concussion did so because they experienced resurfacing of a pre-existing psychiatric condition following a stressful life event (the concussion) or some other unrelated health issue.”

This point is well taken. We have amended our discussion and focus on this limitation on page 15-17 of the discussion section. Specifically, we discuss that our results are restricted to individuals diagnosed in an ER or a hospital setting as well as to non-hospitalized individuals (pages 16-17), who were diagnosed in the ER but were not admitted. Thus, individuals diagnosed by a GP might differ from the population studied here.

In addition, and since our data did not include past psychiatric history, we cannot exclude that exposure and control groups differed in health-related aspects and that these differences biased our results, i.e. that an individual left the workforce for a concussion-unrelated cause like a psychiatric disease triggered by the stress of a concussion event. We discuss, however, why we believe that our quasi-experimental setup leaves us with exposure and control groups that only differ in the timing of concussion, and that both Figure 2 and Tables S1 and S2 show strong support for this assumption. Arguably, because everyone in the control group experiences a concussion within five years after individuals in the exposure group experienced their concussions, the groups are likely to be balanced on unobservable characteristics, as also evidence by the above-mentioned figure and tables.